



# Uncertainty in aerosol-cloud radiative forcing is driven by clean conditions

Edward Gryspeerdt[1], Adam C. Povey[2], Roy G. Grainger[2], Otto Hasekamp[3], N. Christina Hsu[4], Jane P. Mulcahy[5], Andrew M. Sayer[6,7], and Armin Sorooshian[8]

[1]Grantham Institute - Climate Change and the Environment, Imperial College London, London SW7 2AJ, United Kingdom
[2]National Centre for Earth Observation, Department of Physics, University of Oxford, Oxford, OX1 3PU, United Kingdom
[3]Netherlands Institute for Space Research (SRON, NWO-I), Utrecht, The Netherlands
[4]Climate and Radiation Laboratory, NASA Goddard Space Flight Center, Greenbelt, USA
[5]Met Office, Exeter, EX1 3PB, UK
[6]Ocean Ecology Laboratory, Goddard Space Flight Center, National Aeronautics and Space Administration, Greenbelt, MD, USA
[7]GESTAR II, University of Maryland Baltimore County, Baltimore, MD, USA
[8]Department of Chemical and Environmental Engineering, University of Arizona, Tucson, Arizona, USA

**Correspondence:** Edward Gryspeerdt (e.gryspeerdt@imperial.ac.uk)

**Abstract.** Atmospheric aerosols and their impact on cloud properties remain the largest uncertainty in the human forcing of the climate system. By increasing the concentration of cloud droplets ($N_d$), aerosols reduce droplet size and increase the reflectivity of clouds (a negative radiative forcing). Central to this climate impact is the susceptibility of cloud droplet number to aerosol ($\beta$), the diversity of which explains much of the variation in radiative forcing in global climate models. This has

made measuring $\beta$ a key target for developing observational constraints of the aerosol forcing.

While the aerosol burden of the clean, pre-industrial atmosphere has been demonstrated as a key uncertainty for the aerosol forcing, here we show that the behaviour of clouds under these clean conditions is of equal importance for understanding the spread in radiative forcing estimates between models and observations. This means that the uncertainty in the aerosol impact on clouds is, counterintuitively, driven by situations with little aerosol. Discarding clean conditions produces a close agreement

between different model and observational estimates of the cloud response to aerosol, but does not provide a strong constraint on the radiative forcing from aerosol-cloud interactions. This makes constraining aerosol behaviour in clean conditions an important goal for future observational studies.

## 1 Introduction

Changes in atmospheric aerosols driven by human activity can increase the supply of cloud condensation nuclei (CCN; An-

dreae and Rosenfeld, 2008). This increases the cloud droplet number concentration ($N_d$), resulting in smaller droplets and more reflective clouds for a given liquid water content (the Twomey effect; Twomey, 1974), leading to the radiative forcing from aerosol-cloud interactions (RFaci). Changes to the droplet size can have further impacts on the cloud, known as cloud adjustments, including increases in cloud fraction and changes to the liquid water path (Albrecht, 1989). The impact of an-





thropogenic aerosol on these cloud processes is the most uncertain component of the anthropogenic radiative forcing (Bellouin

et al., 2020).

There is a large diversity in the strength of these effects in global climate models (Zelinka et al., 2014; Gryspeerdt et al., 2020), emphasising the need for observation-based estimates of the RFaci and the radiative forcing from adjustments. Given the non-linearity of the $N_d$ response to aerosol, the pre-industrial aerosol burden is a significant uncertainty in these estimates (Carslaw et al., 2013). Pre-industrial conditions are rare in the present day atmosphere (Hamilton et al., 2014), so this aspect is

difficult to constrain with observations. Much of the remaining uncertainty is driven by the activation term ($\beta$), the sensitivity of $N_d$ to an aerosol proxy, $A$ (Quaas et al., 2020).

$$\beta = \frac{d \ln N_d}{d \ln A} \tag{1}$$

The variation in $\beta$ is responsible for much of the variation in aerosol-cloud radiative forcing in ensembles of climate models (Quaas et al., 2009). Recent observational estimates have thus focussed on better estimates of this term. Theoretical consider-ations (Twomey, 1959), supported by observational studies (Gryspeerdt and Stier, 2012; Jia et al., 2021, 2022), suggest it is a

strong function of cloud updraught, entrainment, and hence cloud type.

Previous studies have placed the value for $\beta$ between 0.3 and 0.8, with considerable variation across the globe (Bellouin et al., 2020). This corresponds to a range for the RFaci of between -0.3 and -1.1 W m$^{-2}$. More recent studies have found higher values for $\beta$ and hence more negative values for the RFaci (McCoy et al., 2017; Hasekamp et al., 2019). This variation in $\beta$ explains around 50 % of the uncertainty in RFaci estimates. As cloud adjustments also depend on changes in $N_d$, $\beta$ is also

central to the strength of cloud adjustments. Perfect knowledge of $\beta$ would reduce the uncertainty in the total effective radiative forcing from aerosol (including adjustments) by up to 20 %, based on Bellouin et al. (2020).

The wide range of possible values for $\beta$ limits the accuracy with which the RFaci can be constrained by observations, leading to a wide spread in the RFaci estimates from global climate models. This work demonstrates that the behaviour of clouds in clean conditions (clean clouds) is responsible for much of the diversity in both model and observational estimates of the RFaci.

Reconciling these estimates, this work provides more stringent constraints on the RFaci, and highlights directions for future research.

## 2   Results

### 2.1   Aerosol-$N_d$ relationships

Multiple satellite studies have investigated the link between $N_d$ and an aerosol proxy, such the aerosol optical depth (AOD;

Quaas et al., 2006, 2008), aerosol index (AI; AOD times Ångström exponent; Nakajima et al., 2001; Gryspeerdt et al., 2017), reanalysis aerosol (McCoy et al., 2017) or a retrieval of the CCN (Hasekamp et al., 2019). While they show $N_d$ increasing with aerosol under almost all conditions, the shape of the relationship is typically non-linear. At high aerosol, a saturation effect





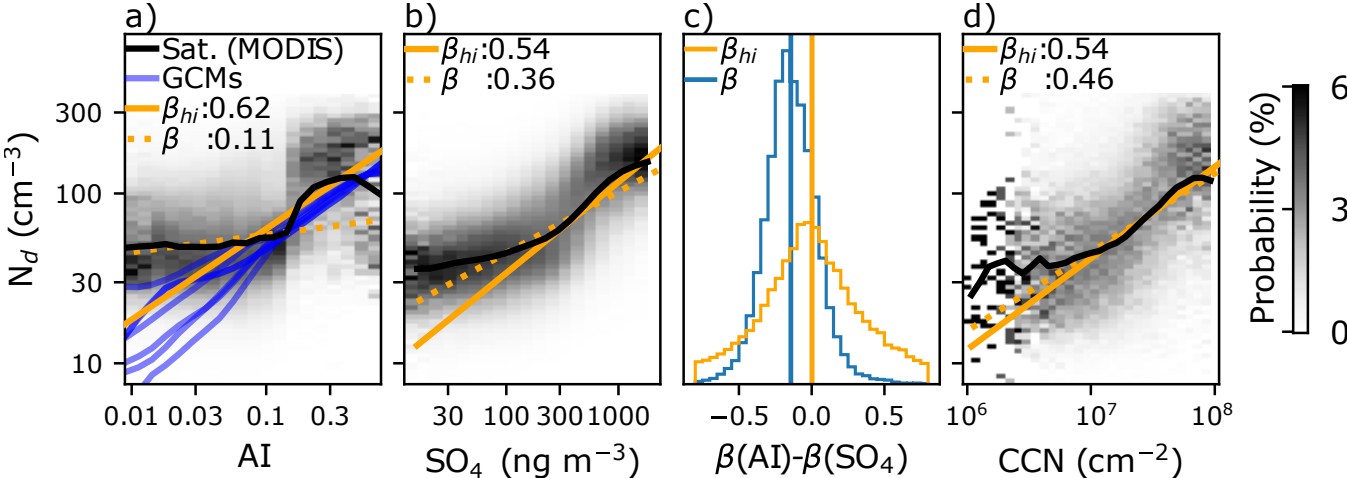

**Figure 1.** a) The relationship between MODIS droplet number concentration ($N_d$) and aerosol index (AI) in the southeast Pacific as a normalised histogram (shading), with the mean value in black. Blue lines are the mean relation for a selection of global aerosol-climate models. The solid orange line is the susceptibility ($\beta$; Eq. 1) fit for data with an AI>0.1, the dashed line for all data. b) as (a) but for reanalysis $SO_4$. c) The global distribution of the difference in $\beta$ for all conditions (blue) and in only polluted conditions ($\beta_{hi}$; orange) calculated with AI and reanalysis $SO_4$, with vertical lines at the global arithmetic mean. d) as (a) but a global mean calculated using retrieved cloud condensation nuclei (CCN) column (following Hasekamp et al., 2019, see Methods).

is sometimes observed (Gryspeerdt et al., 2016; Hasekamp et al., 2019), where the $N_d$ no longer increases with increasing aerosol. This is consistent with a transition to an updraught-limited state (Reutter et al., 2009) and is reproduced in global

model parametrisations of aerosol activation (Fig. 4a, see Methods).

At low aerosol, a reduced $\beta$ is often observed (Fig. 1a, b). The strength of this "shallowing" of the relationship varies with aerosol proxy and is often stronger for observed satellite aerosol proxies (e.g. AOD, AI) than for reanalysis products ($SO_4$). A weaker $\beta$ under low aerosol conditions has also been observed for CALIPSO observations, resulting in an underestimate of the RFaci (Ma et al., 2018).

The "shallowing" effect is responsible for much of the variation in $\beta$ estimates between different aerosol proxies. The two examples shown in Figs. 1a and b use satellite retrieved AI and reanalysis aerosol as the aerosol proxy for a region in the south East Pacific. While they have very different values of $\beta$, they are remarkably similar when considering only high aerosol conditions ($\beta_{hi}$). Similar behaviour was observed in Hasekamp et al. (2019), where $\beta$ (and RFaci) calculated with AI and satellite-retrieved CCN were found to be more similar when low aerosol conditions were excluded.

This convergence of $\beta_{hi}$ is not a purely regional effect. $\beta$ calculated using AI is almost always smaller than when calculated with reanalysis aerosol (Fig. 1c, blue line), resulting in lower RFaci estimates. There is a larger spread when using $\beta_{hi}$, but the global mean difference between the two aerosol proxies moves close to zero and the difference in the global mean value is less than 0.01 ((Fig. 1c, orange line).





The similarity between $\beta_{hi}$ (and variation in $\beta$) calculated using different aerosol proxies demonstrates the importance of clean conditions to this metric. While the observational studies agree on the behaviour of polluted clouds, clean clouds are very common, with over 80 % of retrievals in many locations being in this shallow-$\beta$ regime. These clean clouds thus have a vital role to play in determining the magnitude of $\beta$ and the RFaci.

## 2.2 $\beta$ in global climate models

Quaas et al. (2009) demonstrated that the RFaci in global climate models is a strong function of $\beta$, a relationship that still holds in the latest generation of models (Fig. 2a), with variation in $\beta$ explaining about 50 % of the variance in the RFaci for the models from the AeroCom indirect effect experiment (Zhang et al., 2016; Ghan et al., 2016). Despite several years of progress, the variation in modelled $\beta$ remains high, ranging from 0.25 to 1.5, outside the range identified from satellite studies.

In contrast, global models agree fairly closely on the value for $\beta_{hi}$, which in turn agrees with the value from satellite observations in the range 0.5-0.6 (Fig. 2a). This is a remarkable agreement as although models are regularly assessed using $\beta$, $\beta_{hi}$ is rarely calculated (Ma et al., 2018). Unfortunately, despite this model-observation agreement, $\beta_{hi}$ only provides a weak constraint on the RFaci, with a coefficient of determination of only 0.18 (Fig. 2a). If $\beta$ could be constrained to the same accuracy as $\beta_{hi}$, this could reduce the uncertainty in RFaci by half.

## 2.3 Limits on observing $\beta$

The uncertainty in the satellite-derived values of $\beta$ comes primarily from limitations in the systems that retrieve aerosol properties from space. Satellites use top-of-atmosphere radiances together with assumed microphysical properties to retrieve the optical properties of aerosol and hence infer the physical and chemical properties that are required to calculate CCN concentrations. To do this, they have to separate the aerosol signal from the surface reflectance. Almost all algorithms make assumptions about the spectral and/or directional behaviour of surface reflectance (Kaufman et al., 2002; Sayer et al., 2010) even when using multi-angle polarimeters such as POLDER (Dubovik et al., 2019). Uncertainties in these assumptions are magnified when the aerosol optical signal is small (e.g. under clean conditions).

Under polluted conditions, the aerosol signal is large relative to the surface, such that the key uncertainties are in the aerosol optical properties (single scattering albedo, phase function). In contrast, under clean conditions, the aerosol optical signal is small in comparison to errors in the surface model, increasing the relative error in the retrieved aerosol properties. The larger relative error in the aerosol retrieval under clean conditions reduces the correlation between the CCN and the retrieved aerosol due to regression dilution (Pitkänen et al., 2016). This reduces the magnitude of $\beta$ under clean conditions, as observed in Fig. 1a,b. This issue is particularly severe for AI, which is calculated using the ratio of aerosol optical depths at two wavelengths, resulting in a relative error which tends to infinity (and a $\beta$ which approaches zero) under clean conditions (Fig. 1a).

This explains the larger $\beta$ observed at high sensor zenith angles (Fig. 3). At high sensor zenith angles, the aerosol signal is enhanced relative to the surface signal due to the longer atmospheric path length. The enhanced aerosol signal and correspond-





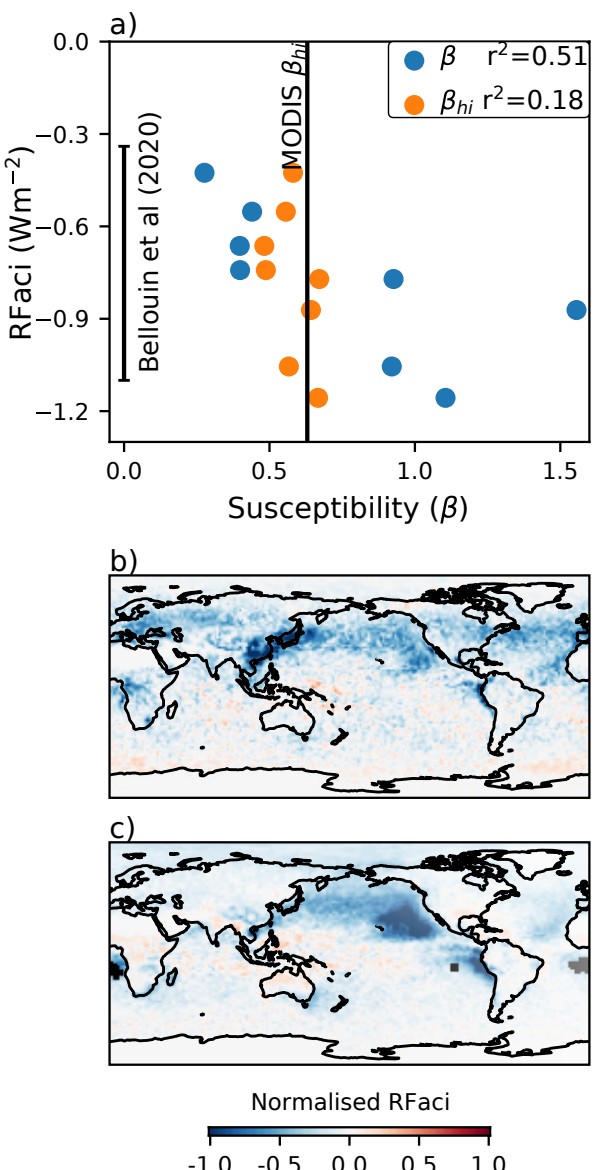

**Figure 2.** a) The relationship between the radiative forcing from aerosol-cloud interactions Gryspeerdt et al. (RFaci; calculated following 2020), susceptibility ($\beta$) in all conditions (blue) and polluted conditions only ($\beta_{hi}$; orange). The MODIS value for $\beta_{hi}$ and the RFaci range are given, following Bellouin et al. (2020). b) and c) the RFaci forcing patterns from ECHAM6-HAM2.2 and CAM5.3-CLUBB-MG2 respectively (see. Ghan et al., 2016), normalised to the same total radiative forcing.

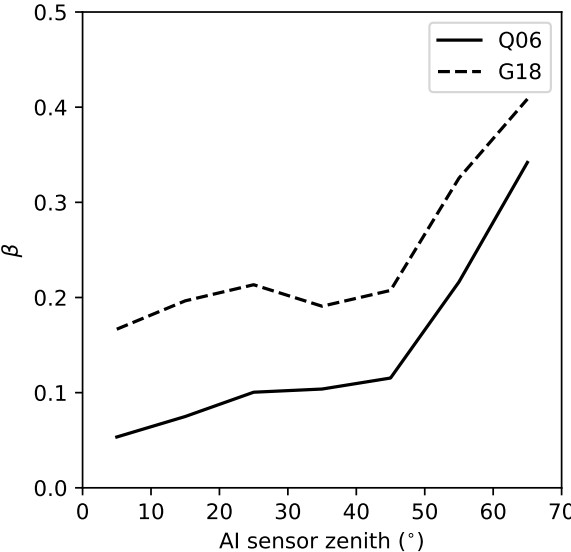

**Figure 3.** Susceptibility ($\beta$) as a function of MODIS sensor zenith angle for the aerosol index (AI) retrieval for two different droplet number concentration ($N_d$) sampling strategies - Q06 (based on Quaas et al., 2006) and G18 (based on Grosvenor et al., 2018) in the south-east Pacific. Note that different sensors are used for the $N_d$ and AI retrievals so that the $N_d$ retrieval is close to nadir, minimising error in the retrieval (Grosvenor et al., 2018; Gryspeerdt et al., 2022).

ing lower effective noise floor (although not necessarily in a straightforward manner; Grosvenor et al., 2018) produces a larger $\beta$, highlighting the importance of the sensitivity of aerosol retrievals under clean conditions to the magnitude of $\beta$.

## 3 Discussion

These results show that $\beta$ under clean conditions is driving diversity in $\beta$ and hence RFaci in both observation-based estimates
and global models. Narrowing the cause of the $\beta$ diversity to the behaviour under clean conditions also enables a better constraint on its magnitude.

Previous observation-based studies have attempted to constrain $\beta$ in clean conditions by assuming the $\beta_{hi}$ is a good approximation for $\beta$, either implicitly by using a satellite simulator (Ma et al., 2018) or explicitly (Hasekamp et al., 2019). However, the aerosol-$N_d$ relationship is non-linear and $\beta_{hi}$ is not necessarily a good guide for the value of $\beta$.

If the aerosol properties remain constant with loading, $\beta_{hi}$ is always equal to or smaller than $\beta$ in clean conditions (Fig. 4a). With lower aerosol loadings being more likely to be aerosol-limited (Fig. 4b), this would suggest that $\beta_{hi}$ is more likely to be an underestimate and could be considered a lower bound for $\beta$ (and hence implied RFaci).

Similar to observation-based studies, some models show a $\beta$ that is lower than $\beta_{hi}$. All of the four models with this behaviour have a minimum $N_d$ implemented in their cloud scheme. This removes any aerosol-$N_d$ dependence at low aerosol, setting $\beta$ for

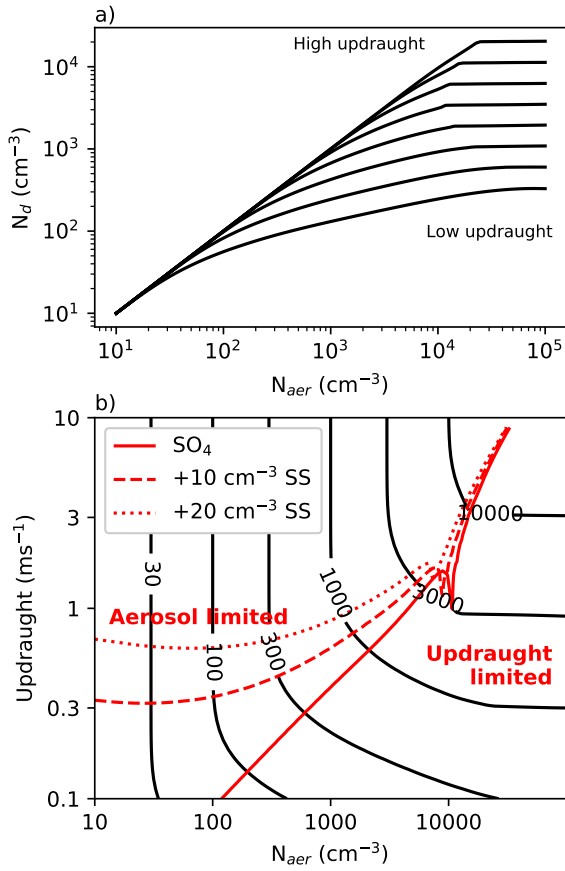

**Figure 4.** a) The relationship between aerosol number ($N_{aer}$) and droplet number concentration ($N_d$) for a selection of updraught speeds. Black lines are $N_d$ contours. The aerosol is a monodisperse ammonium sulphate with a mean radius of 40 nm. b) The $N_d$ as a function of aerosol number and updraught speed for the same monodisperse aerosol. Red contours show the boundary between aerosol-limited and updraught-limited regions. Dashed and dotted red contours show the location of this boundary with the introduction of 10 and 20 cm$^{-3}$ for 850 nm background sea salt (SS) aerosol.

these clean clouds to zero. Sharp changes in cloud properties under very clean conditions (Koren et al., 2014) and the formation of shiptracks in otherwise cloud-free scenes (Gryspeerdt et al., 2021) hint at aerosol limited conditions and a potential minimum $N_d$ for cloud formation. However, the increasing $\beta$ with increasing zenith angle (and sensitivity to aerosol in clean conditions; Fig. 3) suggests that a minimum $N_d$ is not responsible for the low $\beta$. The rarity of low $N_d$ measurements (fewer than 1% below 20 cm$^{-3}$ in Gryspeerdt et al., 2022) suggests that a minimum $N_d$ has little impact on $\beta$ and that $\beta_{hi}$ provides a lower bound for
$\beta$ (and hence the RFaci).





A plausible upper limit for $\beta$ is one, such that increasing aerosol by 10 % increases $N_d$ by 10 %. However, this assumes that the aerosol has constant properties with variations in loading, an assumption that rarely holds in the atmosphere. The addition of even a small amount of coarse sea salt aerosol has the potential to shift low $N_d$ cases from aerosol-limited to updraught-limited conditions, particularly for cases with an updraught less than $1 \, \text{m s}^{-1}$ (typical of stratocumulus, Fig. 4b).

Changes in aerosol type between clean and polluted conditions can also produce $\beta$ values considerably greater than one in global models (Fig. 2a). This effect becomes particularly strong when using an aerosol proxy that depends on the optical properties of the aerosol. This allows $\beta$ values larger than one - further measurements are required to constrain the lower (most negative) bound to the RFaci.

Following Fig. 1c, this would put a likely range on the RFaci of -0.45 to $-1.2 \, \text{W m}^{-2}$, with a more certain upper bound. While this reasoning can constrain the global mean forcing, there is significant regional variation in the pattern of forcing between climate models (e.g. Fig 2b, c). To further reduce this global and regional uncertainty, new retrievals of aerosol properties are required that are accurate at low aerosol loadings, enabling a closer constraint of $\beta$.

## 4   Conclusions

The aerosol impact on clouds through changes in the $N_d$ is one of the largest uncertainties in the anthropogenic forcing of the climate. The diverse observation-based estimates for susceptibility of $N_d$ to aerosol perturbations ($\beta$) is responsible for about half the uncertainty in the RFaci. This diversity comes from clean conditions, where the aerosol optical signal is weak compared to the surface background. This increases the relative error in the retrieved aerosol properties, reducing the magnitude of the calculated $\beta$ through regression dilution (Fig. 1a).

When considering only polluted conditions ($\beta_{hi}$), observational estimates typically agree closely with each other (Fig. 1) and with global models (Fig. 2a). However, although these $\beta$ estimates can be reconciled through $\beta_{hi}$, $\beta_{hi}$ alone is a poor constraint on the RFaci (Fig. 2a). Better constraints on $\beta$, and hence an improved understanding the behaviour of clean clouds, is thus vital to reduce our uncertainty in the RFaci.

Identifying the source of this diversity in RFaci estimates enables a new, stronger constraint on the RFaci (more negative than $-0.45 \, \text{W m}^{-2}$). The lower bound on the RFaci is harder to constrain, as variations in aerosol type can produce $\beta$ values larger than 1 in global models (Fig. 4).

To improve our observation- and model-based estimates of RFaci and address the considerable regional variation (Fig. 2b,c), better retrievals of aerosol behaviour in clean environments are essential. These are challenging for current instruments, but a combination of sensitive in-situ and ground-based remote sensing, together with new satellite instruments and reanalysis data provides a path forward to produce strong constraints on the behaviour of clean clouds and the RFaci.





## Appendix A: Methods

The MODIS AI is used to provide an observational constraint on the RFaci by generating AI-$N_d$ joint histograms from observations. For these histograms, the $N_d$ is calculated using the adiabatic approximation, as specified in Gryspeerdt et al. (2016). The AI is calculated from the AOD-Angström exponent joint histogram in the MODIS MYD08_D3 product using only gridboxes where no ice cloud is detected (to reduce possible cirrus contamination). This stringent filtering for ice clouds also reduces the impact of undetected thin cirrus on the liquid cloud property retrievals (where they are not detected by the multi-layer identification algorithm).

The droplet number concentration ($N_d$) in Fig. 1 was calculated using the adiabatic assumption (Quaas et al., 2006) and the BR17 sampling strategy outlined in Bennartz and Rausch (2017); Gryspeerdt et al. (2022). The G18 sampling strategy (based on Grosvenor et al., 2018) shows similar results, but produces more variable $\beta_{hi}$ values (Fig. 1c). $\beta$ itself was calculated using a linear regression on all available datapoints. This produces a slightly different value for $\beta$ in Fig. 1d when compared to Hasekamp et al. (2019), but the consistent methodology between aerosol proxies simplifies the comparison in Fig. 1.

For Fig. 3, both the Q06 (based on Quaas et al., 2006) and G18 strategies were used to increase data volume. This requires a small sensor zenith angle ($N_d$ is retrieved only close to nadir), so to examine the impact of sensor zenith angle on $\beta$, two MODIS sensors are used, with $N_d$ data from Aqua MODIS and aerosol data from Terra MODIS.

Activated droplet number concentrations are calculated using the parcel model from Rothenberg and Wang (2016), using a mean SO$_4$ radius of 40nm and a sea salt radius of 850nm. Updraught limited conditions are defined as where

$$\frac{d\ln N_d}{d\ln w} > \frac{d\ln N_d}{d\ln N_a}$$

and aerosol-limited cases the converse.

Model data is from the AeroCom indirect effect experiment (Zhang et al., 2016; Ghan et al., 2016), but with UKESM1 replacing HadGEM3-UKCA in the ensemble, due to issues with the HadGEM3-UKCA output. $\beta$ values for UKESM1 are calculated using AOD instead of AI, but this makes little difference to the $\beta$ values (which are very similar to HadGEM3-UKCA). Only gridboxes with an ice water path of less than $1 \text{gm}^{-2}$ and a cloud fraction $>10\%$ are used for the $\beta$ calculation. $N_d$ is converted to an in-cloud value before calculating $\beta$. The RFaci is diagnosed following Gryspeerdt et al. (2020).

*Code and data availability.* The MODIS data was obtained through the Level 1 and Atmosphere Archive and Distribution System (LAADS). The gridded $N_d$ data was obtained through the Centre for Environmental Data Analysis (CEDA). The model data was provided through the AeroCom initiative (http://aerocom.met.no). MERRA-2 data was obtained from the Goddard Earth Sciences Data and Information Services Center (GES DISC). Model data was obtained through the AeroCom initiative (http://aerocom.met.no)

*Competing interests.* At least one of the (co-)authors is an editor of Atmospheric Chemistry and Physics. The authors have no other conflicts of interest.





*Author contributions.* EG, AP and RG designed the study, EG performed the analysis, all of the authors assisted in interpreting the results
175    and writing the paper.

*Acknowledgements.* The authors would like to thank the groups that submitted simulations to the AeroCom model intercomparison project.
This work benefited from discussions within the group "Are we doing the right satellite observations and analyses for quantifying cloud-
mediated aerosol climate forcing?", hosted by the International Space Science Institute (ISSI). EG was supported by a Royal Society Uni-
versity Research Fellowship (URF/R1/191602). AS was supported by NASA grant 80NSSC19K0442 in support of ACTIVATE, a NASA
180    Earth Venture Suborbital-3 (EVS-3) investigation funded by NASA's Earth Science Division and managed through the Earth System Science
Pathfinder Program Office. This study was partly funded through NERC's support of the National Centre for Earth Observation, contract
number PR140015.



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
