# Peer review of "Uncertainty in aerosol-cloud radiative forcing is driven by clean conditions"

_Atmospheric Chemistry and Physics, 2022_

## Author Comment (AC1)

**Reviewer 1**

*: This short letter describes analysis of satellite and global model output data to show that uncertainty in the aerosol impact on clouds is largely driven by situations with little aerosol, i.e., relatively clean conditions. The behavior of clouds in clean conditions is found to be responsible for much of the diversity in both model and observational estimates of the Twomey forcing (radiative forcing from aerosol-cloud interactions, RFaci). This is an important and novel result, which provides another piece of evidence that understanding aerosol-cloud interactions in pristine conditions is important for quantifying aerosol forcing on climate (see e.g., Carslaw et al., 2013; or McCoy et al., 2020). As such, I strongly recommend publication as a letter in ACP. I provide some comments and suggestions for revision, but in my view the manuscript requires only modest changes.*

*The cloud droplet concentration susceptibility to aerosol (beta = -dlnNd/dlnA) is typically established in observations by correlating cloud droplet concentration with an observed aerosol parameter (AOD, AI, CCN, etc. . . ). Observations in the paper, and in some previous studies, indicate that the relationship between Nd and A is not well explained using a simple power law. The relationship tends to flatten for high A, where there are physical arguments for saturation based on droplet activation theory. Here, however, the authors demonstrate here that the Nd-A relationship also tends to flatten for low A, a result which is shown to hold regardless of whether A is represented by aerosol index (AI), sulfate mass loading from reanalysis, or an observational proxy for CCN concentration.*

*The paper demonstrates that if beta is determined using only relatively polluted aerosol levels, then there is excellent agreement between models (beta exhibits only weak intermodel spread). This gives some confidence that the models are accurately representing Nd variability and its dependence on aerosol when aerosol loadings are relatively high. However, if the entire range of A is considered, the values of beta exhibit a wider spread, but the full range beta is far better correlated with RFaci in the models, indicating the importance of being able to accurately quantify the Nd-A relationships across the entire range of A. The wider spread of model-derived beta values indicates greater intermodel uncertainty and therefore a relatively poor constraint on RFaci. The observational estimates of beta for different choices of the A variable also disagree more strongly when the full range of A is considered, which may indicate problems constraining aerosol properties in clean air masses. A variety of reasons could help explain this, including low signal to noise ratio in aerosol retrievals for low A. Future global aerosol measurements will need to embrace the need to function well at lower aerosol loadings than they currently do.*

**Reply**: We thank the reviewer for their comments, which we address below. Line numbers refer to the diff version.

*: My only significant comment pertains to the distinction between ERFaci (Twomey + adjustments) and RFaci (Twomey only, no adjustments). This*

*distinction should be clarified early on. Since only the term "radiative forcing" (RFaci) is used throughout the manuscript, I am assuming that the model simulations used in this study do not include adjustments to aerosol. This distinction is quite important, because recent studies appear to suggest that adjustments, and especially cloud cover adjustments, may be carrying a large fraction of the effective radiative forcing (Chen et al., 2022). Other studies/models show weaker adjustments. So, although beta is perhaps the leading source of uncertainty in RFaci, it is not clear that it is as important for ERFaci. The authors provide a comment on this on line 34-36, no quantitative correlation between susceptibility and ERFaci is provided to firmly establish this.*

**Reply**: We thank the reviewers for this important point. The majority of this work indeed focuses on the RFaci, rather than the adjustments. Previous work has shown that the forcing from the adjustments is proportional to the RFaci in an individual model, but this proportionality varies between models, due to variation in the cloud response to $N_d$ perturbations (Gryspeerdt et al, ACP, 2020).

As most cloud adjustments proceed via a modification of the $N_d$, this means that the ERFaci also depends on $\beta$ (Bellouin et al., 2020). While the extra terms in the adjustments (e.g. the cloud fraction response to $N_d$) are vitally important for accurately determining the ERFaci, a better estimate of $\beta$ is essential for accurate estimates of cloud adjustments (not just the RFaci).

Following the uncertainty estimates in Bellouin et al (2020), a perfect estimate of $\beta$ would reduce the uncertainty in the RFaci by about 50% and the ERFaci by about 20%. We also note that adjustments are typically stronger in clean (low $N_d$) clouds (e.g. Gryspeerdt et al, JGR, 2016), such that the actual impact on the adjustment terms could be larger.

The model simulations used did include adjustments (see Zhang et al, ACP, 2016), but that the ERFaci was decomposed following Gryspeerdt et al (2020) to isolate the forcing from the RFaci alone. This was shown to closely match the RFaci determined using a simulation without cloud adjustments and the RFaci derived using partial radiative perturbations (PRP; Mülmenstädt et al, 2019).

The abstract has been modified to be more explicit about the RFaci focus. The description of the uncertainty contributions for RFaci and ERFaci have also been modified to improve readability (L22, L38). A sentence noting the RFaci/ERFaci distinction has also been included in the conclusions L144.
* * *
*: The distinction seems again to be blurred in Line 4: "...the diversity of which explains much of the variation in radiative forcing in global climate models." This suggests that role for cloud adjustments in driving variation in radiative forcing in models is small, or is that irrelevant here because radiative forcing does not include cloud adjustments?*

**Reply**: Many thanks for pointing this out. This was intended to be a slightly more easily understandable term for the abstract. It has been amended to "radiative forcing from aerosol-cloud interactions" as a correct wording following the results shown in this work.
* * *
***Line 21:*** *Zelinka et al. (2014) examines ERFaci, not RFaci. Is this distinction important?*
**Reply**: This was included here as an important paper that has previously looked at the (E)RFaci from global models. The previous paragraph and following sentence both refer to adjustments (as well as the RFaci), so we prefer to leave it in here.

**Reviewer 2**

*: The manuscript highlights that the elusive sensitivity of Nd to aerosol (Beta) accounts for a significant level of variability in the modeled aerosol-cloud-induced radiative-forcing. Beta from both model and satellite-derived parameters give widely different values. Clean conditions are responsible for much of the diversity of beta values in both models and empirical estimates of the aerosol cloud-induced radiative forcing. The authors show the low aerosol signal/noise at low aerosol loadings limits our ability to constrain Beta. I only have one general comment/suggestion, which I wonder if the authors could elaborate on. I recommend publication of this paper upon minor revisions.*
**Reply**: We thank the reviewer for their comments which we address below. Line numbers refer to the diff version.

**General comments:**

*: "To further reduce this global and regional uncertainty, new retrievals of aerosol properties are required that are accurate at low aerosol loadings, enabling a closer constraint of $\beta$."*

*Can this statement be elaborated on? The authors clearly indicated why Beta is a vital parameter to constrain and identified why it is difficult. Still, it would be conducive if they could provide more insight on what is needed to better constrain Beta other than a general statement such as "new retrievals of aerosol properties are required." For example, are the necessary improvements in remote retrieval accuracy realistic when the relative error approaches infinity for clean cases? I wonder if an accuracy metric could be plotted as a function of the relative error (with the current level of accuracy noted) at various aerosol conditions (clean to polluted) to determine the improvement in our accuracy of aerosol retrievals is needed. This could indicate if the accuracy level needs to be doubled/increased by an order of magnitude/or even more. Perhaps we could produce a beta_medium if the aerosol retrieval sensitivity was improved. Still, it's unclear if the necessary improve the accuracy of aerosol retrievals is possible (at low aerosol loadings) to constrain Beta_all_conditions. If the required improvement in remote retrieval accuracy is unreasonable (for the foreseeable several decades?), perhaps more in-situ measurements are the more promising method to constrain the aerosol loading?*

**Reply**: This is indeed an important point worth mentioning. The exact improvement required is difficult to quantify, but there are some extra bits of information.

Around 44% of retrievals globally are in the "shallow-$\beta$" regime, with an AI<0.1 (now noted at line 70). If this threshold represents an approximate noise threshold, halving this threshold (approximately doubling the retrieval sensitivity) would leave only 20% of retrievals below the threshold and halving again would leave only 5% of retrievals below the threshold. This would provide a significant increase in accuracy for estimating $\beta$ and the RFaci. The increase in $\beta$ at approximately 60 degrees sensor zenith may indicate the impact of doubling the sensitivity (as it has approximately double the atmospheric path length).

The text has now been modified to include this in the discussion at line 131.

**Specific comment:**
* * *
**Line 74::** *remove "as"*
**Reply**: This sentence has been amended.